

# Brief communication: Hydrologic connectivity of a tidewater glacier characterized with Sentinel-2 satellite images – a case study of Nordenskiöldbreen, Svalbard

Jan Kavan [1,2] and Vincent Haagmans [3,4]

[1]Polar-Geo-Lab, Department of Geography, Faculty of Science, Masaryk University, Brno, CZ
[2]Alfred Jahn Cold Region Research Centre, Institute of Geography and Regional Development, University of Wroclaw, Wroclaw, PO
[3]WSL Institute for Snow and Avalanche Research SLF, Davos Dorf, CH
[4]Institute of Environmental Engineering, ETH Zürich, Zürich, CH

**Correspondence:** Jan Kavan (jan.kavan.cb@gmail.com)

**Abstract.** Direct measurements of glacier hydrological processes are usually restricted to short periods and a limited number of sites due to logistical, financial, and meteorological constraints. As a result, the indirect study of glacier hydrology through remote sensing has gained traction while the accessibility of high-resolution publicly available remote sensing data has also increased. By quantifying the areal extents of key dynamic features of a tidewater glacier (i.e. the evolution of sea ice, supraglacial lakes, meltwater plumes) as proxies of its hydrological cycle using Sentinel-2 observations, a simple alternative amidst the outlined logistical constraints is potentially available. Here we demonstrate the usefulness of Sentinel-2 satellite images as a simple and accessible tool with high temporal coverage for studying glacier hydrology. To make this case for the Nordenskiöld tidewater glacier, the evolution of its supraglacial hydrological system and respective meltwater plumes areal extents were monitored for the 2016-2020 melting seasons. Hydrological connectivity of supra- and subglacial systems and the resulting meltwater plumes are illustrated. Meltwater is stored on the glacier surface at the beginning of the melt season (June) which is observed through the filling of the supraglacial lakes. The stored meltwater is later released (June/July), probably through englacial conduits and moulins, and consequently reaches the subglacial drainage system. The resulting occurrence of meltwater plumes clearly indicates the latter in Adolfbukta at the glacier terminus. This signals the transport of significant volumes of water in contact with the glacier bed. The meltwater plume activity peaks during late July, and its appearance continues until mid-September. The duration of the glacier melt season is reflected through the filling of supraglacial lakes and later in the appearance of meltwater plumes. The temporal pattern of the hydrologic processes is relatively uniform during the study period, contrasting the large variability of sea ice cover duration. The observed behavior of Nordenskiöld's supraglacial lakes is in good agreement with similar tidewater glaciers in Svalbard.

## 1 Introduction

Glaciers globally fulfill a fundamental role in the water cycle through the temporal storage of water and its partial release during a relatively short melt period. In the Arctic, the seasonal melt period is generally restricted to a few spring and summer





months. Surface meltwater in glaciated areas with few or no drainage pathways is commonly collected in supraglacial lakes. These lakes are drained when an efficient connection to the subglacial drainage system is established (Veen, 2007). Meltwater is then routed through a complex (combined) network of lakes, supraglacial, englacial, and subglacial channels. Onward,
subglacial runoff is usually the leading process of draining glacier meltwater. This behavior has been described for different glaciers and ice caps around the Arctic (e.g., Chu et al., 2009; Das et al., 2008). If a glacier is of the marine-terminating type (i.e., tidewater), subglacial runoff can potentially be observed through the occurrence of (sediment-laden) meltwater plumes at the glacier front. Moreover, glacier hydrology and ice motion are intricately linked through the reduction of basal traction by meltwater accessing the glacier sole (Irvine-Fynn et al., 2011). Such complex hydrologic networks often drain the tidewater
glaciers through massive meltwater plumes with significant amounts of suspended sediments, which can be easily distinguished in the marine environment.

Direct measurements of glacier hydrologic processes are usually restricted to short periods and a limited number of sites due to financial, meteorological (Leidman et al., 2020), and logistical constraints (Alexander et al., 2020). In response, the indirect study of glacier hydrology and runoff production through remote sensing has gained traction during recent years (e.g.,
Narama et al., 2017; Jouvet et al., 2018; Yang et al., 2019; Bunce et al., 2021; Turton et al., 2021; Kavan and Haagmans, 2021). Moreover, recent advances in the accessibility of publicly available remote sensing data with high temporal and spatial resolutions (e.g. Sentinel Hub EO Browser) have supported such endeavors. At the same time, the need for expert knowledge to process this abundance of remotely sensed data has been limited. Hence can one, without expert knowledge of processing remote sensing data, use Sentinel-2 imagery to study a subject as dynamic as the hydrological system of a tidewater glacier?
By quantifying the areal extents of key dynamic features (i.e., evolution of sea ice, supraglacial lakes, meltwater plumes) as proxies of glacier hydrology from Sentinel-2 observations, a simple alternative amidst the outlined constraints is potentially available. In order to test the latter, the hydrologic regime of the tidewater glacier Nordenskiöldbreen, Svalbard, during the 2016-20 melt seasons served as a case study. Note that less than 20% of the approximately 1100 glaciers on the Svalbard high-Arctic archipelago are of the tidewater type (Błaszczyk et al., 2009). Nevertheless, marine areas where such glaciers meet
the sea are often considered biological hotspots, directly providing the adjacent marine environment with essential nutrients and other mineral materials. Moreover, through the rising of already sediment-laden nutrient-rich subglacial meltwater plumes towards the surface, large volumes of additional ambient nutrient-rich deep seawater are dragged upwards (Meire et al., 2017). As a result, the occurrence of certain animal species, typically marine mammals and birds in Svalbard, is tightly related to this specific ecosystem (Lydersen et al., 2014). Tidewater glacier is, for example, representing a unique pupping habitat for
harbor seals (Womble et al., 2021). The latter underlines the relevance of closely monitoring these environments with simple and accessible tools. Even more so when considering that Svalbard experienced an average air temperature warming rate that was seven times greater than the global average since 1991 (Nordli et al., 2020) and that Svalbard glaciers continue to retreat significantly (Geyman et al., 2022) also when compared to other tidewater glaciers in the Northern hemisphere (Kochtitzky and Copland, 2022).



## 2   Nordenskiöldbreen

Nordenskiöldbreen originates west off the vast Lomonosov ice cap in central Spitsbergen, Svalbard. Flowing around two nunataks, Terrierfjellet and Ferrierfjellet, it descends steeply to terminate in Adolfbukta, a north-eastern branch of Billefjorden (Fig. 1). Nordenskiöldbreen is the only calving glacier in this basin and has an approximately 3.5 km wide partial calving front (Ewertowski et al., 2016). For at least a significant part of the 20th century, the glacier front was known to calve along its entire width (Plassen et al., 2004) and since has retreated 35 m.a$^{-1}$ on average (Rachlewicz et al., 2007). However, the extent of the calving front may become more significant again in the upcoming decades since the upstream bedrock surface is below sea-level, as revealed by a GPR survey (Pelt et al., 2012). Presently, the southern and northern margins and the center of the glacier front (on Retreat Island) are land-based, as derived from satellite imagery through the absence of crevasses. Observed morphological features in the terrestrial forelands indicate the occurrence of cold-based ice at the lateral margins of Nordenskiöldbreen and warm-based conditions close to the central flow line (Ewertowski et al., 2016). Hence, the glacier is considered polythermal (Hagen et al., 1993).

Nordenskiöldbreen covers an area of 193 km$^2$, has ice thicknesses exceeding 600 m (Pelt et al., 2012), and surface flow velocities along the main flow line in the range of 30-60 m.a$^{-1}$ (den Ouden et al., 2010). For the period 1989-2010, van Pelt et al. (2012) reported the equilibrium line altitude at 719 m a.s.l. and an overall negative mass balance was determined (-0.39 m w.e. year$^{-1}$). A less negative net glacier mass balance of -0.09 m w.e. year$^{-1}$ was estimated by Gustavsson (2019) for the period 2005-2017. Nordenskiöldbreen is classified as a non-surging glacier type (Błaszczyk et al., 2009; Ewertowski et al., 2016). However, note that some morphological structures on the southern foreland indicate potential surge behavior prior to observations starting in 1882 (Ewertowski et al., 2016). The input of subglacial sediment from Nordenskiöldbreen to the marine environment was studied in 2001 and 2002 by Szczuciński and Zajączkowski (2012). Significant vertical particulate matter deposition rates were reported during this period, i.e., several thousand g m$^{-2}$ day$^{-1}$ during the short summer season and tens of g m$^{-2}$ day$^{-1}$ in the autumn. In turn, these suspended sediments were important for sedimentary coast formation at the nearby Kapp Napier (Kavan, 2020).

Furthermore, Nordenskiöldbreen has its accumulation zone on the Lomonosov ice cap and has been subject to a multitude of studies, including mass balance and GPR surveys (van Pelt et al., 2012), firn properties, and water percolation processes (Marchenko et al., 2017), and climate reconstruction from ice cores (Isaksson et al., 2001; Vega et al., 2015). The frontal zone of the glacier with the immediately adjacent areas was also frequently studied, especially for its geomorphological processes (Ewertowski et al., 2016), coastal development (Strzelecki, 2011), or terrain formation after the glacier retreat (Allaart et al., 2018). The retreating glacier is also interesting in terms of its biological processes: Pinseel et al. (2017) studied the diatom communities in lakes developed after glacier retreat, whereas Vinšová et al. (2015) also focused on communities in the cryoconite holes on the glacier surface similarly to Vonnahme et al. (2016).

The local climate is characterized as continental dry with low winter temperatures and high summer temperatures in comparison to the west coast of Svalbard. Low winter temperatures and the distance from the open sea are expressed in the extended sea ice cover duration, typically from December to May/June (Nilsen et al., 2008). However, this may change with the ongoing



increase of air and seawater temperatures, and the related decline of sea ice cover around Svalbard (Nordli et al., 2020). The
90 mean annual air temperature in the adjacent Petuniabukta ranges from -3.7°C near the coast to -8.4°C on top of the Mumien
Peak (773 m a.s.l.) (Ambrožová and Láska, 2017). Mean monthly air temperatures exceeding 0°C were recorded from June to
September and reached a maximum of 15°C in July (Láska et al., 2012). During the study period (1 January 2016 – 26 August
2020), the mean air temperature was -3.5 °C.

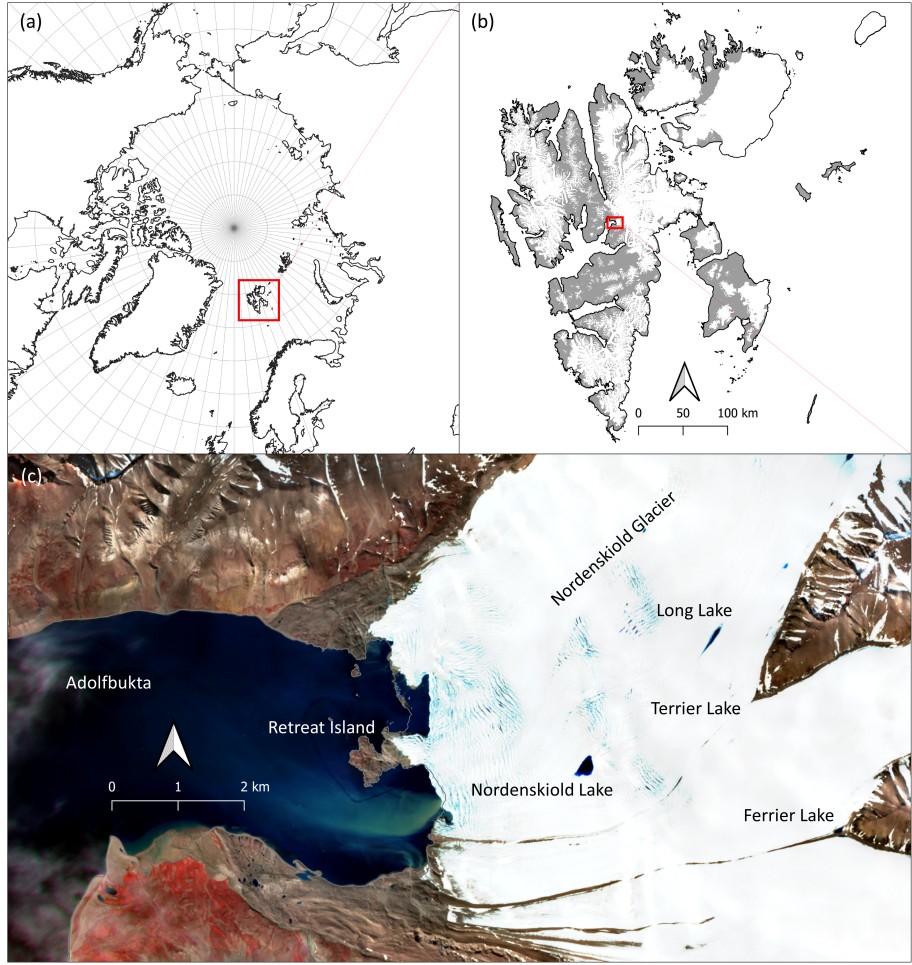

**Figure 1.** (a) location of the Svalbard archipelago within the Arctic, (b) location of Adolfbukta with Nordenskiöldbreen on the Svalbard
archipelago, and (c) Sentinel-2 satellite image of Adolfbukta (image taken on 30 July 2020) with names of relevant topographic features.



## 3 Material and Methods

### 3.1 Remote sensing

The areal extents of sea ice, sediment-laden meltwater plumes, and selected supraglacial lakes were quantified using Sentinel-2 satellite false-color images (bands 8,4,3) obtained from the Sentinel Hub EO Browser (https://www.sentinel-hub.com/). Images covering the assumed ablation season (15 May – 30 September) with cloud coverage smaller than 20% were extracted and checked for their suitability. The resulting selection of satellite images was then used for assessing the areal extent of meltwater plumes and supraglacial lakes. Images from the spring season (1 April – 30 June) were used to quantify sea ice extent. Respectively, 15 images for 2016, 13 for 2017, 24 for 2018, 25 for 2019, and 24 images for 2020 were obtained (see list of dated images in Supplementary material). Figure 2 presents an exemplary Sentinel-2 satellite image sequence. The images were downloaded as georeferenced TIFF files and processed in QGIS software by manually delimiting the areal extent of studied features (supraglacial lakes, sediment plumes, and sea ice extent). Areal extent of each feature was delimited for each image to obtain the timeseries of features studied. The area of such delimited shapes was then generated automatically in the QGIS software and used for further analyses.

### 3.2 Atmospheric forcings

Mean daily air temperature observations from the automatic weather station (AWS) located on the western coast of Petuniabukta (N78.71060°; E16.46059°) were used. This AWS sits at an altitude of 14 m a.s.l. and approximately 10 km from the Nordenskiöldbreen front. Here, an EMS33H sensor at the standard height of 2 m above the surface recorded air temperatures. The data set was complemented by air temperature observations from Adventdalen (50 km southward) for the periods with missing observations (19 July 2019 – 17 August 2019; 27 August 2020 – 30 September 2020). The two data sets were checked for their correlation: a correlation coefficient of 0.94 was determined for the mean daily air temperature records. Incoming radiation would be a second-order factor affecting the melt rate. However, unlike the air temperature which is relatively spatially homogeneous, the radiation is strongly affected by cloud cover, which is site-specific; therefore, observed radiation from Petuniabukta was not used.

## 4 Results and Discussion

### 4.1 Sea ice

During the early melt season, sea ice is often reported as a constraint for monitoring meltwater plumes (e.g., Chu et al., 2009; McGrath et al., 2010; Dowdeswell et al., 2015; Schild et al., 2017). However, this does not seem to be the case for Nordenskiöldbreen as the break up of sea ice in Adolfbukta occurred well before the onset of glacier surface melt and first signs of meltwater plumes. The seasonal decline of sea ice in Adolfbukta from full cover to an ice-free surface is characterized by a relatively short breakup period of about 2-3 weeks (Fig. 3c). This process usually occurred between the end of May and



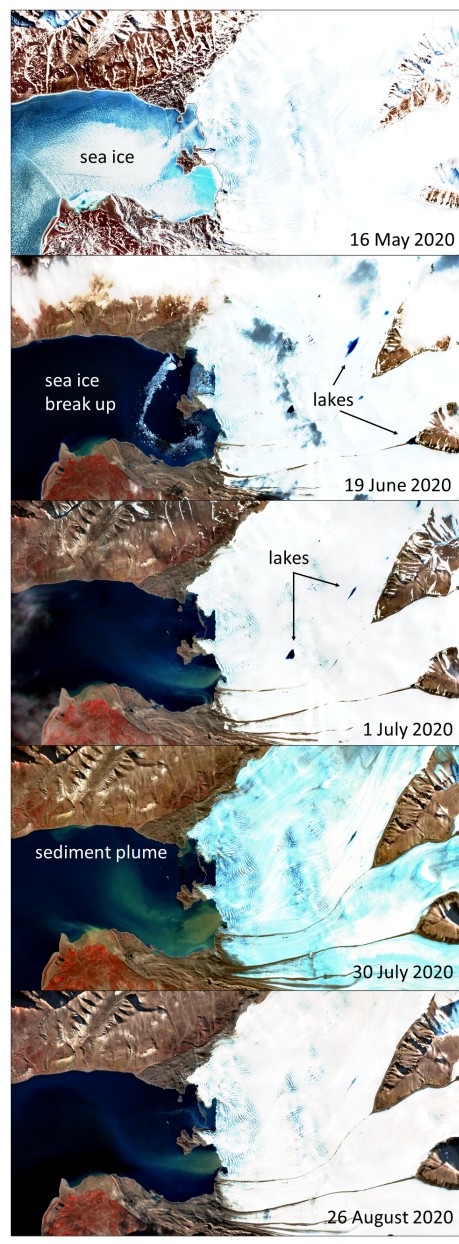

**Figure 2.** A typical sequence of Sentinel-2 images of the 2020 spring and summer season showing casing the temporal evolution of sea ice cover, supraglacial lakes, and melt water plume areal extents.

the beginning of June, with Adolfbukta having losing its sea-ice by mid-June. The only exception was the year 2016, when the

breakup of sea ice occurred about a month earlier than usual, leading to a completely ice-free Adolfbukta by mid-April 2016.





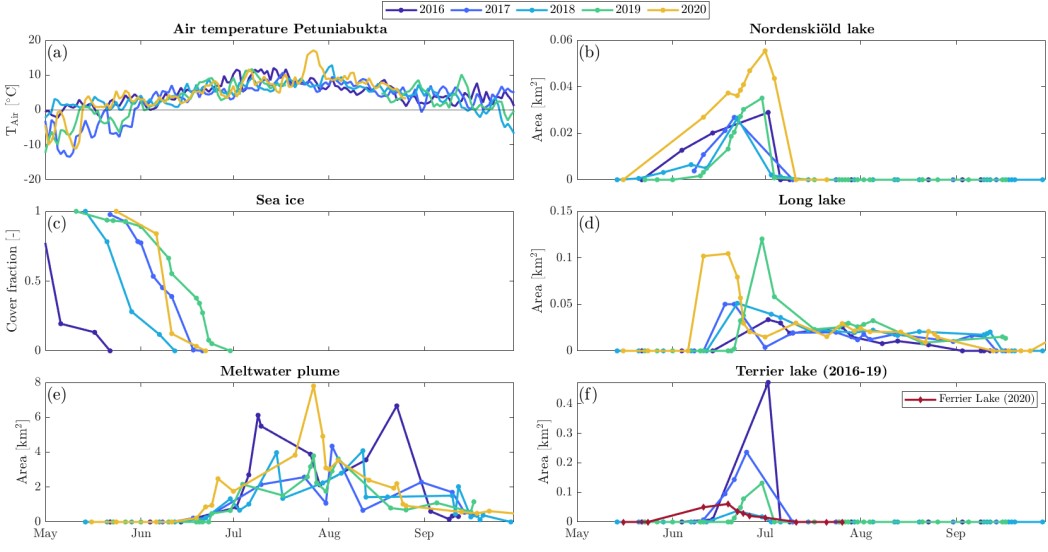

**Figure 3.** Overview of observed seasonal variability during the 2016-20 study period of (a) air temperature observed in Petuniabukta near Nordenskiöldbreen, (b) Nordenskiöld supraglacial lake areal extents (c) sea-ice areal extents in Adolfbukta, (d) Long lake supraglacial areal extents, (e) meltwater plume areal extents in Adolfbukta, (e) Terrier supraglacial lake areal extents, (f) Terrier lake and Ferrier lake supraglacial lake areal extents.

## 4.2 Supraglacial lakes

Filling of the supraglacial lakes on Nordenskiöldbreen occurs during early summer. While the filling of the Nordenskiöld Lake and Terrier Lake was gradual, their subsequent drainage was rather abrupt. Probably due to the opening up of the subglacial conduits. The Nordenskiöld Lake (see Fig. 1) starts to fill late May and reaches its maximum extent in late June or the beginning

of July (Fig. 3b). A similar pattern was observed for Terrier Lake, with the filling phase taking place about two weeks later. However, the drainage of Terrier Lake coincided with the drainage of Nordenskiöld Lake and is dated to the beginning of July. Note that the Terrier Lake extent was minimal and that Ferrier Lake only appeared in 2020 for the first time (red curve in Fig. 3f). Both latter lakes are positioned at the downstream boundary where the nunataks meet the glacier surface and fill the depressions where the central moraines form. The surface topography of the glacier suggests that these depressions were

filled with meltwater potentially originating from snow stored on the slopes of these nunataks and the glacier surface near the lake basin. Ferrier Lake filled and drained rather gradually, albeit earlier in the season when compared to Terrier Lake. All studied supraglacial lakes revealed relatively simultaneous filling and drainage. This suggests either a hydrological connection through the englacial and subglacial channel network or separate channel networks but at a similar stage of development. How et al. (2017) reported similar findings: observed lakes were clustered into groups with temporally similar behavior. Other

than the Nordenskiöld supraglacial lakes, the three lake clusters observed on Kronebreen had different timing of filling and





drainage.The filling of two supraglacial lake clusters was shifted towards the peak melt season (July/August), while the third cluster had a similar temporal pattern through its filling during June and draining in late June/July.

The filling and draining of Nordenskiöld's supraglacial Long Lake exhibited a slightly different pattern. Unlike the other supraglacial lakes studied, Long Lake did not completely drain during the melt season (Fig. 3D). The maximum lake areal

extent was reached simultaneously with the other supraglacial lakes in late June and the beginning of July. Consequently, it drained only partially. The lake area decreased gradually from mid-July to mid-September when it drained, probably because of the ceasing meltwater supply. Whereas Schild et al. (2016) reported the onset of sediment plumes before the supraglacial lakes' drainage, the Nordenskiöldbreen supraglacial lakes drainage precedes the onset of sediment plumes (Fig. 3-4). Note that in addition to the above studied supraglacial lakes, numerous much smaller lakes occur on the surface of Nordenskiöldbreen

during the melting season. However, their area is generally so small that this inherently leads to significant area delineation error.

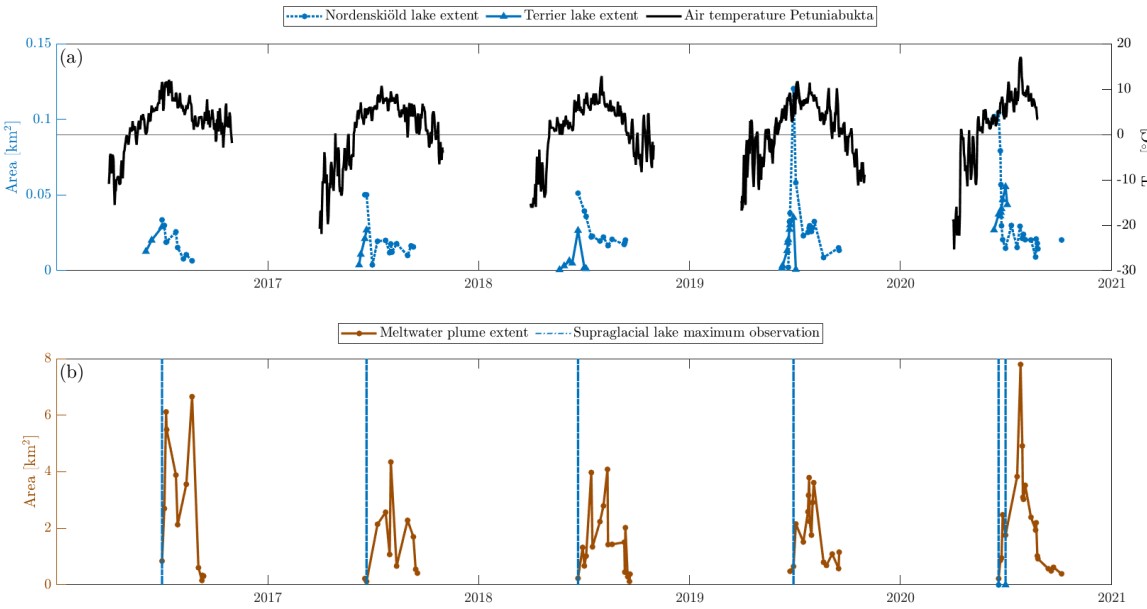

**Figure 4.** (a) Observed extents of two supraglacial lakes on Nordenskiöldbreen relative to local mean daily air temperatures for which the horizontal line represents the zero-degree isotherm. (b) Extents of the observed meltwater plumes and vertical lines representing maximum supraglacial lake extents. Clearly, supraglacial lake maximums are followed by increases in meltwater plume extents.

### 4.3    Meltwater plumes

The areal extent of meltwater plumes exhibited similar temporal patterns throughout the 5-year study period (Fig. 3). First signs of subglacial runoff into the fjord were recorded during the second half of June, while the peak activity was identified during





July and August. The subglacial transport of suspended sediments ceased during September, indicated by the lack of meltwater plumes and hence runoff. The observed meltwater plumes originated from the southern part of the glacier front and exhibited a stable spatial pattern. This is supported by turbidity observations dated to July 2019 along the glacier front by Szeligowska et al. (2021), which were significantly higher close to the southern part. Moreover, inter-annual variability of the areal plume extent was limited. The three single largest areal plume extents were recorded in 2016 and 2020. However, these observations are

affected by the temporal coverage of the selected satellite images. They do not necessarily indicate that suspended sediments' transport was actually significantly higher during both years. This is also the Achilles heel of studying glacier hydrology through satellite remote sensing using Sentinel-2 since it does not allow for sub-daily observations. Moreover, this approach is also dependent on visual coverage of the study area uninhibited by cloud cover. A workaround would be the use of local time-lapse photography (e.g., Schild et al., 2016), which inherently demands fieldwork and the annual service of deployed

sensors.

The period of active subglacial drainage, expressed through the presence of proglacial meltwater plumes combined with the timing of their maximum areal extent, is in good accordance with Kronebreen in NW Svalbard (How et al., 2017; Schild et al., 2017) and Tunabreen in central Svalbard (Schild et al., 2017). The input of suspended sediment material from Kronebreen during the two months of 2015 summer melting, quantified at approximately $2.48 \pm 0.37 \times 10^6$ tons (Meslard et al., 2018), is

even higher than similar estimates of Greenlandic tidewater glaciers of comparable basin size. A somewhat more prolonged period of meltwater plume activity (beginning of June to the end of September) was observed in Kangerlussuaq Fjord, West Greenland, for non-tidewater glaciers. In addition, McGrath et al. (2010) reported a tight correlation between areal meltwater plume extent and discharge from the ice sheet, suggesting the potential of using meltwater plume extent as a proxy for meltwater discharge. Further, more northward in West Greenland meltwater plumes were observed from the beginning of June until the

first half of September (Schild et al., 2016). The meltwater plumes in the surface layer in front of Nordenskiöldbreen exhibited the same temporal pattern as Hansbreen (south Svalbard). Nevertheless, significant amounts of suspended sediments were observed even during autumn in the deeper zones of the latter fjord (Moskalik et al., 2018), underlying a high local process variability.

The subglacial drainage of marine-terminating glaciers is essential for the local environment. Large amounts of fresh water

and sediments are transported from land to sea, including high concentrations of nutrients. For example, Meire et al. (2017) identified increased primary production at the front of a tidewater glacier in Greenland during the period of meltwater plume activity (July, August). Subglacial discharge of the Greenland tidewater glaciers was also identified as essential for the entrainment of nutrients from deeper seawater towards the surface and thus enriching the local marine environment with positive feedback on phytoplankton blooms (Hopwood et al., 2018). Upward movement of the subglacial plumes amplifies the general

circulation of water at the glacier front. In Svalbard, these processes were crucial for creating life hotspots for birds and marine mammals (Lydersen et al., 2014). The fjord's geometry and the plume outlet depth's influence the vertical stratification and movement of water in the fjord and near the glacier terminus (Carroll et al., 2015). In the case of Nordenskiöldbreen, the meltwater plume outlet is situated close to the sea surface. At the same time, the depth of the adjacent fjord is just a few tens of meters at maximum depth which would probably restrict the formation of any vertical movement of water. Vonnahme et al.





(2021) found evidence for upwelling movement at the southern part of the Nordenskiöldbreen terminus, despite the shallow waters. Such inflow was as well found during the spring period when the adjacent area is covered by sea ice, even though the flow was considerably lower than during the summer period (Vonnahme et al., 2021). We argue that spring inflow of meltwater originated probably without contact with the subglacial environment as no visual evidence of sediment inflow was observed. Finally, van Pelt et al. (2018) also found a short-lived impact of summer melt on ice flow during the cold season and identified

the runoff and glacier flow velocity as highly variable and season dependent.

### 4.4 Atmospheric forcings

The observed air temperature pattern is relatively uniform in all years studied (Fig. 3a). Generally, air temperatures exceed the freezing point during May and stay above 0°C until mid-September. The maximum values of approximately 10°C were observed during late July. This period coincided with the vastest meltwater plumes recorded during the study period. The

highest recorded mean daily air temperature (17.1°C) occurred on 27 July 2020, the day with the maximum observed meltwater plume extent. To further investigate this observation, the set of selected Sentinel-2 images was studied to define of a potential short-term statistical relation between air temperature and meltwater plume activity. The correlation between air temperature and sediment plumes area is illustrated in Figure 5, which summarizes all the available sediment plume areal extent data from the five years studied. It is no surprise that a positive correlation was found. The best correlation was found in the case of

aggregated five consecutive days' average air temperature preceding the sediment plumes occurrence. This suggests the large inertia of the glacier system and correspondingly delayed reaction.

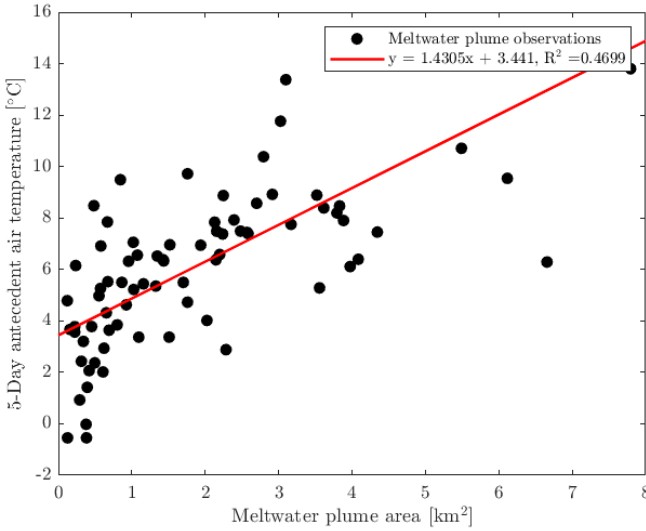

**Figure 5.** Relationship between observed meltwater plume areal extents and five days antecedent average air temperature. The red line represents a robust simple linear regression as a correlation measure using the Theil-Sen estimation which is less sensitive to outliers.



## 4.5 Conceptual model

A conceptual model illustrated in Figure 6 captures the Nordenskiöldbreen seasonal hydrological system dynamics and includes all studied processes: the breakup of the sea ice is the first sign of a response to increasing air temperatures and radiation. This is accompanied by the onset of snowmelt on the glacier surface and expressed through the filling of supraglacial lakes in the frontal zone of the glacier. Consequent abrupt draining of these lakes suggests activation of the glacier's englacial and subglacial drainage pathways. The latter is promptly accompanied by massive subglacial runoff expressed through vast meltwater plumes visible on the surface of Adolfbukta near the subglacial drainage outlet in the southern part of the glacier front. The latter behavior repeated itself during all studied years with only minor variations and indicates high stability of the glacier drainage system. The meltwater plumes activity ceases during September with the air temperature getting down to zero or below zero in the upper parts of the glacier.

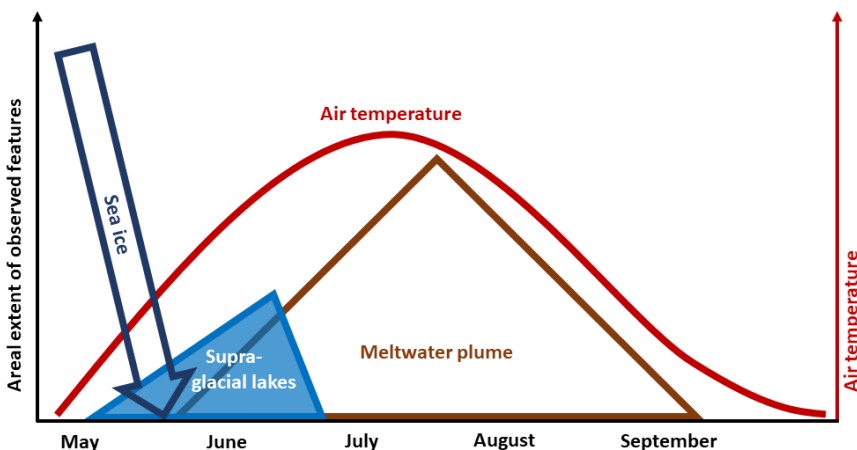

**Figure 6.** Conceptual model of the Nordenskiöldbreen hydrological network functioning. Sea ice break up during May/June followed by increasing areal extent of supraglacial lakes with its abrupt drainage (end of June); drainage of supraglacial lakes coincides with the increase in areal extent of meltwater plumes. The meltwater plumes activity ceases during September. All these processes are driven by air temperature dynamics as a main factor influencing meltwater production

## 5 Conclusions

We demonstrated that simple optical remote sensing imagery is potentially a valuable tool to describe the hydrologic connectivity of a tidewater glacier. Comparing the five consecutive seasons revealed a repetitive spatio-temporal pattern of processes connected to the hydrologic system of the glacier. The supraglacial lakes in the frontal ablation part of the glacier were filled with water originating from melting the surface snow cover during mid to late June. This water was rapidly released later with the opening of the englacial drainage system and activation of the subglacial channels. The rather abrupt drainage of

the supraglacial lakes at the beginning of the summer season activated the whole hydrologic system of the glacier. As a result, the water entering the subglacial environment was in contact with the glacier bed loaded by fine-grained sediments. This

was exhibited by the presence of vast sediment-laden meltwater plumes in front of the glacier terminus, i.e., on the surface of Adolfbukta, and detectable on Sentinel-2 images. The subglacial drainage system of the glacier stayed active during the summer season till mid or late September, when the occurrence of meltwater sediment plumes ceased. The magnitude of the sediment plumes recorded depended on the atmospheric conditions affecting the meltwater generation and was generally highest in late July or the beginning of August. It was shown that a series of consecutive warm days lead to the maximum areal

extent of plumes observed. A great example is the absolute maximum plume area recorded on 27 July 2020, while on this day also the absolute maximum average daily air temperature was recorded for the study period (17.1°C).

*Data availability.* The remote sensing data used for the study are publicly available through the Sentinel Hub EO browser (https://apps.sentinel-hub.com/eo-browser). Air temperature data from Adventdalen are provided by UNIS (https://www.unis.no/resources/weather-stations), the air temperature data from Petuniabukta are available upon request.

*Author contributions.* JK designed the study, led data analysis, and writing. VH assisted in the data analysis, designed the figures, and contributed to writing. Both authors approved the final paper.

*Competing interests.* The authors declare that they have no conflict of interest.

*Acknowledgements.* This work was supported by the Masaryk university project ARCTOS MU (MUNI/G/1540/2019) and Arctic Field Grant project 'Glacier mass balance – central Svalbard' (RiS ID 10903), funded by The Research Council of Norway. The research was

also funded through the Norwegian Financial Mechanism 2014–2021: SVELTA—Svalbard Delta Systems Under Warming Climate (UMO-2020/37/K/ST10/02852) based at the University of Wroclaw. Air temperature data from Petuniabukta were obtained thanks to CzechPolar Project (LM2015078) and provided by Kamil Láska. These were complemented by air temperature data from Adventdalen provided by UNIS. The logistic support of the staff of the Adam Mickiewicz University Polar Station is highly appreciated as well.





**Table 1.** Observation dates of Sentinel-2 images used for the study.

| 2016 | 2017 | 2018 | 2019 | 2020 |
|------|------|------|------|------|
| 6-4-2016 | 22-5-2017 | 14-5-2018 | 11-5-2019 | 24-5-2020 |
| 20-4-2016 | 27-5-2017 | 21-5-2018 | 21-5-2019 | 6-6-2020 |
| 28-4-2016 | 31-5-2017 | 29-5-2018 | 23-5-2019 | 11-6-2020 |
| 30-4-2016 | 1-6-2017 | 7-6-2018 | 27-5-2019 | 19-6-2020 |
| 6-5-2016 | 5-6-2017 | 12-6-2018 | 1-6-2019 | 22-6-2020 |
| 17-5-2016 | 8-6-2017 | 22-6-2018 | 10-6-2019 | 24-6-2020 |
| 22-5-2016 | 11-6-2017 | 30-6-2018 | 11-6-2019 | 26-6-2020 |
| 4-6-2016 | 18-6-2017 | 3-7-2018 | 19-6-2019 | 1-7-2020 |
| 14-6-2016 | 21-6-2017 | 6-7-2018 | 20-6-2019 | 21-7-2020 |
| 2-7-2016 | 10-7-2017 | 15-7-2018 | 21-6-2019 | 27-7-2020 |
| 6-7-2016 | 24-7-2017 | 17-7-2018 | 23-6-2019 | 30-7-2020 |
| 9-7-2016 | 31-7-2017 | 30-7-2018 | 24-6-2019 | 31-7-2020 |
| 10-7-2016 | 2-8-2017 | 5-8-2018 | 30-6-2019 | 1-8-2020 |
| 26-7-2016 | 12-8-2017 | 12-8-2018 | 4-7-2019 | 4-8-2020 |
| 29-7-2016 | 31-8-2017 | 13-8-2018 | 17-7-2019 | 14-8-2020 |
| 13-8-2016 | 10-9-2017 | 20-8-2018 | 25-7-2019 | 22-8-2020 |
| 23-8-2016 | 14-9-2017 | 10-9-2018 | 26-7-2019 | 23-8-2020 |
| 3-9-2016 | 17-9-2017 | 11-9-2018 | 27-7-2019 | 25-8-2020 |
| 9-9-2016 | | 12-9-2018 | 28-7-2019 | 26-8-2020 |
| 10-9-2016 | | 16-9-2018 | 31-7-2019 | 13-9-2020 |
| 12-9-2016 | | 18-9-2018 | 2-8-2019 | 18-9-2020 |
| | | 19-9-2018 | 4-8-2019 | 22-9-2020 |
| | | 20-9-2018 | 21-8-2019 | 6-10-2020 |
| | | 29-9-2018 | 26-8-2019 | |
| | | | 5-9-2019 | |
| | | | 16-9-2019 | |
| | | | 17-9-2019 | |





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
