# Peer review of "Brief communication: Hydrologic connectivity of a tidewater glacier characterized with Sentinel-2 satellite images – a case study of Nordenskiöldbreen, Svalbard"

_The Cryosphere, 2022_

## Referee Comment (RC1)

**Review of "Brief communication: Hydrologic connectivity of a tidewater glacier characterized with Sentinel-2 satellite images – a case study of Nordenskiöldbreen, Svalbard" (tc-2022-54) by Jan Kavan & Vincent Haagmans**

**Summary**

The authors lay out the issue that field measurement of glacier hydrology is difficult, yet significant computing & remote sensing expertise is often needed to ingest the large volumes of freely available satellite data. The authors therefore ask the question, "can one, without expert knowledge of processing remote sensing data, use Sentinel-2 imagery to study a subject as dynamic as the hydrological system of a tidewater glacier?" The authors manually digitize Sentinel imagery to characterize the areal extent of sea ice, supraglacial lakes, and sediment plumes as they evolve over the melt season of several years. The authors demonstrate that Sentinel is a useful tool for glacier hydrology surveillance, but don't really ask or answer any science questions that advance the field other than "how do these things vary?". I don't see any serious flaws in the study, but think the study generally illustrates something that is already well-known (i.e., that satellite data, and particularly high spatial & temporal resolution data, is useful for glacier research). The study could be strengthened by framing their research around a process-based question about glacier hydrology. However, that is something of a value judgement/opinion, and think the manuscript could be publishable after addressing the following major & minor comments, constituting what I think are "minor revisions".

**Major Comments**

1) In Section 2, you lay out a lot of background on previous work conducted on Nordenskiöldbreen, but never state "why this glacier". Why is this your study site? Are there reasons to expect that this is a better/worse site for a Sentinel case study than any other Svalbard glacier (or global glacier, for that matter)?

2) On L96 you state that "the areal extents of sea ice, sediment-laden meltwater plumes, and selected supraglacial lakes were quantified using Sentinel- 2 satellite false-color images" but give no criteria for how you are going to define these features? What image interpretation cues are you using to say something is sea ice vs. icebergs, do you consider frozen supraglacial lakes, etc.?

3) Use of transition words like "furthermore" sometime seem out of place/incorrect. "Furthermore" implies that you're building on a previous argument, but "additionally" seems like a better transition word (you're often just adding more information, not necessarily "building a case") > for example L78.

**Minor Comments**

L24: Need a citation to support that subglacial discharge is generally the primary mechanism for meltwater export.

L43: Perhaps worth noting that the tidewater glaciers disproportionately contribute to Svalvard glacier area/mass flux/mass loss (if true).

L45: O'Neel et al. (2015) would be a good reference to support this statement > doi: 10.1093/biosci/biv027

L53: Svalbard glaciers are retreating significantly faster than other Northern Hemisphere glaciers? If so, "faster" is missing from this sentence.

L63: Unclear > your evidence that the glacier is partly land terminating is that there's no crevasses? Can you not just see land in the images? There are other ways to get crevasse free ice.

L64: Can you provide a little more information about what these "morphological features" are?

L68: Incorrect period after "m" in $30 - 60$ m a$^{-1}$

L98: Surely given the magnitude of previous work done on this glacier, someone has quantified the melt season and you don't have to just assume a start/end date? Or can you use meteorological data to identify when temperatures go above 0 °C on average?

L112: Is there any temperature offset between the two met stations? Do you correct for it?

Fig 3b: How is sea ice concentration defined?

L128: The sentence starting with "probably" should be combined with the previous sentence – it is not a complete sentence.

L134: How do you know the water came from snow on the mountains and isn't locally sourced glacier melt?

L137: Can you see surface streams in the imagery? That could help you rule out surface connection between the lakes accounting for the synchronous timing. I suspect your latter hypothesis (separated systems responding to a shared forcing) is the most likely.

L139: Clustered how? Spatially or by their behavior? This is also discussion, not results.

L163-165: You should note that you can also use microwave remote sensing to see through clouds, coarser resolution traditional satellites (e.g., MODIS), or cubesats (e.g., Planet) to get around these cloud & temporal sampling issues.

L166: Subglacial discharge may not make it to the fjord surface if: 1) the flow is hyperpycnal due to high suspended sediment load, or 2) the upwelling discharge plume reaches neutral buoyancy below the surface (e.g., Donald Slater papers). These limitations should be acknowledged in using plumes as your sole proxy for subglacial discharge. (Recurring at L193).

L189: Citation or more justification needed that the discharge conditions would inhibit vertical water movement.

L198: Strange to say maximum temp is approximately 10 °C and then say 17 °C in the following sentence. Please clarify different metrics or simply statement.

L206: This finding could also suggest that subglacial hydrology acts as a flow integrator over these timescales.

L212: "Massive" is subjective, and you don't actually quantify subglacial discharge > reword.

L231: This is stylistic, but I would suggest the conclusion have a "wrap up" sentence instead of ending with this specific example of air temp-plume area relationship.

Review by William Armstrong

---

## Referee Comment (RC2)

**Review of "Brief communication: Hydrologic connectivity of a tidewater glacier characterized with Sentinel-2 satellite images – a case study of Nordenskiöldbreen, Svalbard"**
**Authors: Jan Kavan, Vincent Haagmans**

**Summary**

The authors present an analysis of the changes in supraglacial lake area, meltwater plume area, and sea ice concentration during the summer seasons of 2016-2020 at Nordenskiöldbreen, Svalbard. By manually digitizing these features from Sentinel-2 imagery, they demonstrate that simple remote sensing approaches can yield valuable information on the hydrological system of a tidewater glacier. The data presented in the manuscript is indeed important to understanding Nordenskiöldbreen and is, I believe, worthy of publication. However, the details of the methods are sometimes unclear, and the manuscript is currently lacking in originality and impact. I agree with reviewer William Armstrong that "the study could be strengthened by framing their research around a process-based question about glacier hydrology." There should be more discussion of the significance of this work and new insights; an explanation of what this work contributes to the field of glacial hydrology or our understanding of tidewater glaciers. Addressing this, clarifying details of the methods, and addressing other comments described below would constitute significant revisions that would help increase the impact of this manuscript.

**General Comments**

1) There should be more discussion of the limitations of this method, both technical limitations relating to the remote sensing method and environmental limitations in your ability to detect plumes. There is some discussion of technical limitations (temporal resolution and cloud cover) in lines 161-165 (in the Results and Discussion section on meltwater plumes), but this should be expanded on and moved to a different section as it is also a limitation for detecting sea ice and supraglacial lakes. A mention of spatial resolution limitations could also be added (difficulty of observing very small features, if they exist).

2) Regarding the limitation of observing plumes using remote sensing, there needs to be some comment on the possibility of plumes existing without surface expression. The distance a plume rises before reaching neutral buoyancy depends on the discharge volume and rate, fjord stratification, and mélange/sea ice presence/rigidity (Carroll et al., 2015; Slater et al., 2015; De Andrés et al., 2020; Everett et al., 2021). Additionally, the organization of the subglacial hydrologic system determines whether discharging water forms a concentrated plume or not (Melton et al., 2022). A channelized subglacial hydrologic system will result in a discharging plume, whereas a distributed system would result in discharging water to be dispersed along the terminus. Therefore, you cannot conclude that there is no discharge when there is no plume, and you also cannot assume that there is no plume when you do not observe a plume at the surface (in this case there could be a plume at depth that reaches neutral buoyancy before reaching the surface.) You must make it clear that your record of plumes is *conservative* and be careful about

making conclusions about what is happening when a surfacing plume is not visible throughout the manuscript.

3) Details about the methods need to be clarified to ensure that the approach is valid and reproducible. Why was the false-color band combination used? How did you define the edge of sediment plumes? How was sea ice cover fraction calculated? In general, how were all the features (lakes, plumes, and sea ice) defined and identified in the imagery? (This relates to reviewer Armstrong's Major Comment 2.)

4) Throughout the manuscript, and especially in the introduction/background, you should consider including additional key references to recent articles about meltwater plumes (e.g., Slater et al., 2018; De Andrés et al., 2020; Cook et al., 2021; Everett et al., 2021; Melton et al., 2022).

5) This is a comment a reviewer once gave me, which also applies here: "As a reader, every time the importance of another study is mentioned by name, it takes the focus away from the amount of work that has gone into this study… A suggested modification would be to summarize the findings of prior research as they relate to this work (instead of who was the author and what they found) and then citing the authors in parentheses at the end of the sentence." In general, you could aim for more parenthetical citations and less narrative citations.

**Specific Comments**

Line 7: Is it appropriate to say that Sentinel-2 images have "high temporal coverage?" Perhaps compared to some other satellite imagery, but you would really need time-lapse imagery to get high enough temporal coverage to see when lakes drain and plumes appear/disappear.

Line 13: The first time you mention Adolfbukta, it would be helpful to clarify that this is the name of the fjord.

Lines 20-25: Some of the sentences in the introductory sentences about glacial hydrology need references (the sentences that do not currently have references).

Lines 32-34: "Direct measurements of glacier hydrologic processes are usually restricted to short periods and a limited number of sites due to financial, meteorological, and logistical constraints. In response, the indirect study of glacier hydrology and runoff production through remote sensing has gained traction"
This is almost the exact same wording as in the abstract. Re-write either this or the abstract to fix this repetition.

Line 37: "At the same time, the need for expert knowledge to process this abundance of remotely sensed data has been limited." This is unclear. Do you mean that the expert knowledge is limited or that there is no need for it? I expect you mean the expert knowledge is limited, so I would suggest deleting "the need for."

Lines 40-42: "By quantifying the areal extents of key dynamic features (i.e., evolution of sea ice, supraglacial lakes, meltwater plumes) as proxies of glacier hydrology from Sentinel-2 observations, a simple alternative amidst the outlined constraints is potentially available." Again, word-to-word this is almost the exact same sentence that appears in the abstract. Re-write to avoid repetition.

Lines 44-46: This needs a reference: "Marine areas where such glaciers meet the sea are often considered biological hotspots, directly providing the adjacent marine environment with essential nutrients and other mineral materials."

Line 55: The Nordenskiöldbreen section reads like a review paper on the glacier. Perhaps it could be consolidated/summarized to include only that information which is directly relevant to this study.

Lines 59-62: You talk about where the glacier calved in the past and allude to the fact that it no longer calves along the entire front. Where does calving happen now? A few sentences describing the current calving behavior would help establish the present-day context. The following sentence touches on this, but it could be elaborated further.

Lines 62-63: Move this sentence before the previous sentence to help establish present context before moving on to the future: "Presently, the southern and northern margins and the center of the glacier front (on Retreat Island) are land-based, as derived from satellite imagery through the absence of crevasses."

Paragraph starting at line 78: This paragraph especially reads like a review of all the studies done on the glacier. Consider consolidating this paragraph and using parenthetical citations (at the end of sentences – see General Comment 5).

Line 90: Describe/define what Petuniabukta is the first time you introduce it.

Line 93: Reference for the mean air temperature during the study period? (Or, if this is part of your results, don't include it in this section.)

Figure 1: When I first saw this figure, it was unclear where Terrier and Ferrier Lakes were (especially because it looks like there is no water in Terrier Lake at this time). Is there some way to label the location of the lakes more clearly, either with arrows or drawn outlines?

Line 97: Why was this false-color band combination chosen? This is a NIR-red-green combination (this should be stated for readers unfamiliar with Sentinel bands), which is commonly used for areas covered in vegetation to visualize plant health. Discuss why this false-color combination was beneficial to your study.

Line 98: How was the "assumed ablation season" determined/decided?

Line 100: How was it decided that only images from the spring season would be used for sea ice extent?

Line 103: How was the boundary of the sediment plume determined? It seems like the visual signature of the plume on the surface of the fjord does not have a clear boundary as the turbid water fades out gradually. Without determining a threshold based on the spectral signature of the turbid water, I'm not sure how a visual estimation of the plume boundary could be standardized.

Line 105: This part of the methods is unclear and needs to be elaborated on: "The area of such delimited shapes was then generated automatically in the QGIS software and used for further analyses." What do you mean by "generated automatically?" Do you mean that the areas of the shapes were calculated in QGIS? (If so, say that instead.)

Methods section: For sea ice, how was the cover fraction defined/calculated? What was the total area used to determine the fraction of this total area that was covered in sea ice? Sea ice cover fraction is not mentioned in the methods, but it is plotted in Figure 3.

Figure 2: The label for sediment plume is not in the location where the plume appears to be coming from. An arrow should be used to indicate where the plume is coming from, or the location of the label could be moved.

Figure 3 caption: Should it read "air temperature observed in Petuniabukta and Adventdalen?"

Lines 128-129: It is suggested that abrupt lake drainage is "probably due to the opening up of subglacial conduits." However, abrupt drainage would be more indicative of opening of a moulin or hydrofracturing through crevasses (e.g., Das et al., 2008; Danielson and Sharp, 2013).

Line 132: Why do you think Ferrier Lake only filled in one year? Do you have a hypothesis for the physical mechanism explaining this?

Line 134: Seconding the comment by reviewer William Armstrong. Why do you propose that the water is from snowmelt?

Lines 141-142: "The filling of two supraglacial lake clusters was shifted towards the peak melt season (July/August), while the third cluster had a similar temporal pattern through its filling during June and draining in late June/July." This level of detail may not be necessary for describing someone else's results from a different glacier.

Line 147: This paper by Shield et al. is focused on Rink Isbræ, West Greenland. While it is good to mention their findings in the background and potentially other parts of the manuscript, I'm not sure it makes sense to mention one result from Greenland so specifically in your results/discussion section. If comparing your results to those of Schild et al., why not compare them to all the other papers studying plumes and supraglacial lake drainage? This could be done more generally, referencing studies together at the end of a sentence (parenthetical citations) instead of within the sentence (narrative citations). (See General Comment 5 and comment above on parenthetical vs. narrative citations.)

Figure 4: Why is Long Lake's extent not included in this figure?

Lines 155-156: "The subglacial transport of suspended sediments ceased during September, indicated by the lack of meltwater plumes and hence runoff." This inference is too far-reaching. You can't say it ceased because sediment-laden subglacial water could still be discharging without creating a surface expression visible in satellite imagery. Subglacial water could be discharging in a more dispersed pattern without forming concentrated plumes (indicating distributed rather than channelized subglacial hydrology), or the plume could still exist without rising enough to create a surface expression. (See General Comment 2 and our discussion in Melton et al., 2022.) Even if you did know for sure that there was no discharge of suspended sediment, there could still be _subglacial_ transport of suspended sediments beneath the glacier. More analysis would be needed to support this conclusion (e.g., a time-series of meltwater discharge/runoff).

Lines 156-157: "The observed meltwater plumes originated from the southern part of the glacier front and exhibited a stable spatial pattern." Does this say anything about the spatial network of meltwater drainage through and under the glacier? Simple subglacial flow routing based on hydraulic potential calculations could add significance to the location of the observed plume and even lake locations. If DEMs of the glacier surface and bed are available for your study area, you could consider including this. See Everett et al. (2016), How et al. (2017), and Melton et al. (2022).

Lines 161-165: The limitations of this method should be discussed in a different section because they are limitations for everything (including sea ice concentration and supraglacial lakes), not just the plumes. See General Comment 1.

Line 172: This needs a reference: "A somewhat more prolonged period of meltwater plume activity (beginning of June to the end of September) was observed in Kangerlussuaq Fjord, West Greenland, for non-tidewater glaciers."

Lines 188-189: "the meltwater plume outlet is situated close to the sea surface." Isn't it at the base of the glacier, not close to the sea surface?

Line 192-193: "We argue that spring inflow of meltwater originated probably without contact with the subglacial environment as no visual evidence of sediment inflow was observed." This does not necessarily mean that the meltwater didn't have contact with the glacier bed. The sediment plume may not have been visible during this time if it did not reach the fjord surface, or subglacial discharge could be dispersed instead of concentrated as a plume. See General Comment 2.

Lines 194-195: This last sentence of the section seems out of place in the results/discussion section without more discussion of ice velocity and/or including velocity data in your study.

Lines 201-202: "the set of selected Sentinel-2 images was studied to define of a potential short-term statistical relation between air temperature and meltwater plume activity." This is confusing wording. You can just say what you did (I think a linear regression between temperature and plume area).

Figure 5 caption: Add a short statement of what the "Theil-Sen estimation" is and a reference.

Lines 222-224: "The rather abrupt drainage of the supraglacial lakes at the beginning of the summer season activated the whole hydrologic system of the glacier." I think this might be too far-reaching. There was likely water present in englacial and subglacial networks before the lake drainage which could have been active, even if it were not discharging. I'm not sure you can say that the lake drainage activated the whole hydrologic system.

Lines 230-231: This last sentence of this manuscript does not belong in the conclusions and is especially out of place as the last sentence of the paper. The manuscript needs another couple of sentences or a paragraph on the broader significance and impact of this work / recommendations for future work.

**Technical Corrections**

Throughout the manuscript, there are several times when "the latter" is used, and it is unclear what this is referring to. The first time I noticed it is in line 13: "… clearly indicates the latter in Adolfbukta at the glacier terminus." I would suggest removing "the latter" here and elsewhere and saying specifically what you are referring to. In most cases this can be fixed by simply changing "the latter" to "this."

Line 42: Unclear use of "the latter"

Line 49: Change "Tidewater glacier is" to "Tidewater glacier termini are" or similar

Line 50: Unclear use of "the latter"

Line 51: Change "Even more so" to "This is even more important" or similar

Line 52: Change "continue" to "continued"

Line 53: Remove "also"

Line 67: Change "has ice" to "with ice"

Lines 83-85: For this sentence, all three references could be at the end of the sentence instead of mixed into the beginning, middle, and end.

Line 101: Written as is, "respectively" is not needed. It could be written: "Respectively, 15, 13, 24, 25, and 24 images were obtained for 2016, 2017, 2018, 2019, and 2020."

Line 102: "(see list of dated images in Supplementary material)": This is labeled as Table 1. You can just say "(Table 1)" or, if it is supposed to be in the Supplementary material, "(Table 1 within the supplementary material)"

Lines 103-105: This is repetitive, saying the same thing in different ways: "manually delimiting the areal extent of studied features (supraglacial lakes, sediment plumes, and sea ice extent). Areal extent of each feature was delimited for each image to obtain the timeseries of features studied." Consolidate these two sentences into one to avoid repetition.

Figure 2 caption: change "showing casing" to "showcasing"

Figure 3: The order of lettering in this figure is a bit confusing. Maybe the letters could increase sequentially down the columns (a,b,c in the first column and d,e,f in the second column) so that the lakes would be d,e,f instead of b,d,f. This would make the caption easier to read.

Figure 3f: The title should indicate both Terrier Lake and Ferrier Lake instead of only Terrier Lake.

Figure 3 caption: There are 2 (e)s in the caption. Delete "(e) Terrier supraglacial lake areal extents"

Line 128: At the beginning of the sentence, change "Probably due to" to "This was probably due to" (or combine the two sentences into one)

Line 139: Change "Other than" to "Different from" or "Unlike"

Line 186: Change "depth's" to "depth"

Line 191: Change "as well" to "also"

Line 198: Change "freezing point" to "melting point"

Line 204: "It is no surprise that a positive correlation was found." This is subjective and unnecessary. This sentence can be deleted.

Lines 204-205: Change "The best correlation was found in the case of aggregated five consecutive days' average air temperature preceding the sediment plumes occurrence" to "The best correlation resulted from averaging the air temperature over the five consecutive days preceding the sediment plumes' occurrences"

Line 210: I suggest changing "onset of snowmelt on the glacier surface" to "onset of surface melting" or similar.

Line 212: Unclear use of "the latter." Change to "This"

Line 213: Same comment as above. Change "The latter" to "This"

Line 215: change "with the air temperature getting down to zero" to "while the air temperature drops to zero"

Line 219: change "consecutive seasons" to "consecutive years" or "consecutive summers" or "consecutive melt seasons"

**References**

Carroll D, Sutherland DA, Shroyer EL, Nash JD, Catania GA and Stearns LA (2015) Modeling turbulent subglacial meltwater plumes: Implications for fjord-scale buoyancy-driven circulation. J. Phys. Oceanogr. 45(8), 2169–2185 (doi:10.1175/JPO-D-15-0033.1)

Cook SJ, Christoffersen P, Truffer M and Chudley TR (2021) Calving of a large Greenlandic tidewater glacier has complex links to meltwater plumes and mélange. J. Geophys. Res. Earth Surf. doi:10.1029/2020JF006051.

Danielson, B., and Sharp, M. (2013). Development and application of a time-lapse photograph analysis method to investigate the link between tidewater glacier flow variations and supraglacial lake drainage events. *J. Glaciol.* 59, 287–302. doi:10.3189/2013JoG12J108.

Das, S. B., Joughin, I., Behn, M. D., Howat, I. M., King, M. A., Lizarralde, D., et al. (2008). Fracture propagation to the base of the Greenland ice sheet during supraglacial lake drainage. *Science.* 320, 778–781. doi:10.1126/science.1153360.

De Andrés E, Slater D, Straneo F, Otero J, Das S and Navarro F (2020) Surface emergence of glacial plumes determined by fjord stratification. *Cryosph.*, 1–41 (doi:10.5194/tc-2019-264)

Everett, A., Murray, T., Selmes, N., Rutt, I. C. C., Luckman, A., James, T. D. D., et al. (2016). Annual down-glacier drainage of lakes and water-filled crevasses at Helheim Glacier, southeast Greenland. *J. Geophys. Res. Earth Surf.* 121, 1819–1833. doi:10.1002/2016JF003831.

Everett A, Murray T, Selmes N, Holland D and Reeve DE (2021) The impacts of a subglacial discharge plume on calving, submarine melting and mélange mass loss at Helheim Glacier, south east Greenland. *Journal of Geophysical Research: Earth Surface,* 126. doi:10.1029/2020JF005910.

How, P., Benn, D. I., Hulton, N. R. J., Hubbard, B., Luckman, A., Sevestre, H., et al. (2017). Rapidly changing subglacial hydrological pathways at a tidewater glacier revealed through simultaneous observations of water pressure, supraglacial lakes, meltwater plumes and surface velocities. *Cryosphere* 11, 2691–2710. doi:10.5194/tc-11-2691-2017.

Melton, S., Alley, R., Anandakrishnan, S., Parizek, B., Shahin, M., Stearns, L., LeWinter, A., and Finnegan, D. (2022) Meltwater drainage and iceberg calving observed in high-spatiotemporal resolution at Helheim Glacier, Greenland. *Journal of Glaciology*. https://doi.org/10.1017/jog.2021.141.

Slater DA, Nienow PW, Cowton TR, Goldberg DN and Sole AJ (2015) Effect of near-terminus subglacial hydrology on tidewater glacier submarine melt rates. *Geophys. Res. Lett.* **42**, 2861–2868 (doi:10.1080/10550887.2011.581988)

Slater, D. A., Straneo, F., Das, S. B., Richards, C. G., Wagner, T. J. W., & Nienow, P. W. (2018). Localized plumes drive front-wide ocean melting of a Greenlandic tidewater glacier. Geophysical Research Letters, 45. https://doi.org/10.1029/2018GL080763.

---

## Author Comment (AC1)

**Reply to William Armstrong (RC1)**

Review of "Brief communication: Hydrologic connectivity of a tidewater glacier characterized with Sentinel-2 satellite images – a case study of Nordenskiöldbreen, Svalbard" (tc-2022-54) by Jan Kavan & Vincent Haagmans

Summary
The authors lay out the issue that field measurement of glacier hydrology is difficult, yet significant computing & remote sensing expertise is often needed to ingest the large volumes of freely available satellite data. The authors therefore ask the question, "can one, without expert knowledge of processing remote sensing data, use Sentinel-2 imagery to study a subject as dynamic as the hydrological system of a tidewater glacier?" The authors manually digitize Sentinel imagery to characterize the areal extent of sea ice, supraglacial lakes, and sediment plumes as they evolve over the melt season of several years. The authors demonstrate that Sentinel is a useful tool for glacier hydrology surveillance, but don't really ask or answer any science questions that advance the field other than "how do these things vary?". I don't see any serious flaws in the study, but think the study generally illustrates something that is already well-known (i.e., that satellite data, and particularly high spatial & temporal resolution data, is useful for glacier research). The study could be strengthened by framing their research around a process-based question about glacier hydrology. However, that is something of a value judgement/opinion, and think the manuscript could be publishable after addressing the following major & minor comments, constituting what I think are "minor revisions".

Major Comments
1) In Section 2, you lay out a lot of background on previous work conducted on Nordenskiöldbreen, but never state "why this glacier". Why is this your study site? Are there reasons to expect that this is a better/worse site for a Sentinel case study than any other Svalbard glacier (or global glacier, for that matter)?
**REPLY:** Good point, this core question was not addressed by the authors. A paragraph has been added at the end of the introduction:
'The tidewater glacier Nordenskiöldbreen was selected as a study subject due to the combined availability of local meteorological records from the shores of an adjacent bay, Petuniabukta, and a significant body of literature focused on the glacier. Moreover, a rather dry continental climate ensures a relatively high number of clear sky days compared to the coastal regions of Svalbard. Hence, a sufficient number of Sentinel-2 images can be used to illustrate the seasonal dynamics of the glacier system. Finally, over a decade of combined local field experience around the study site supports the interpretation of observations.'

2) On L96 you state that "the areal extents of sea ice, sediment-laden meltwater plumes, and selected supraglacial lakes were quantified using Sentinel- 2 satellite false-color images" but give no criteria for how you are going to define these features? What image interpretation cues are you using to say something is sea ice vs. icebergs, do you consider frozen supraglacial lakes, etc.?
**REPLY:** Thanks for this question, it was somehow obvious for us, but certainly not for the reader. We included a more detailed description of how the features were delimited. In fact, both supraglacial lakes and sea ice cover are very contrasting features and easily visually detectable. The supraglacial lakes did not freeze, they all drained before the below zero temperatures arrived in mid September. Also there were no lakes before May/June when they started to fill in with meltwater. In the case of icebergs, there were usually only a few of them incorporated within the sea ice cover. Delimitation of sediment plumes is of course more complicated as the boundary between the plume and clear sea water is not always so clear - in the corrected manuscript we use improved methodology that we tested in a

different location of Svalbard ad which was published recently in Kavan et al. 2022 (10.3390/w14121840)

3) Use of transition words like "furthermore" sometime seem out of place/incorrect. "Furthermore" implies that you're building on a previous argument, but "additionally" seems like a better transition word (you're often just adding more information, not necessarily "building a case") > for example L78.
**REPLY:** The transcript was scanned for similar cases as on L78 and if necessary reworded.

Minor Comments
L24: Need a citation to support that subglacial discharge is generally the primary mechanism for meltwater export.
**REPLY:** We believe that this is supported with the following sentence (also with references).

L43: Perhaps worth noting that the tidewater glaciers disproportionately contribute to Svalbard glacier area/mass flux/mass loss (if true).
**REPLY:** Tidewater glaciers in Svalbard contribute disproportionately to the Svalbard glacier areal cover. Has been added to the manuscript as: 'About 15% of the glaciers on the Svalbard high-Arctic archipelago are of the tidewater type, yet these make up as much as 60\% of the glacier-covered area (Blaszczyk et al., 2009)'

L45: O'Neel et al. (2015) would be a good reference to support this statement > doi: 10.1093/biosci/biv027
**REPLY:** Thank you for the suggestion. It has been included.

L53: Svalbard glaciers are retreating significantly faster than other Northern Hermisphere glaciers? If so, "faster" is missing from this sentence.
**REPLY:** This does not necessarily seems to be the case (e.g. Kochtitzky and Copland, 2022) and therefore 'faster' was not included

L63: Unclear > your evidence that the glacier is partly land terminating is that there's no crevasses? Can you not just see land in the images? There are other ways to get crevasse free ice.
**REPLY:** You are right, we adjusted the sentence to "Presently, the southern and northern margins and the center of the glacier front (on Retreat Island) are land-based, as derived from satellite imagery through the absence of crevasses and also from in situ observation where the bedrock appears beneath the glacier masses.".

L64: Can you provide a little more information about what these "morphological features" are?
**REPLY:** These are mostly ice-cored moraines close to the centerline and fluted surfaces close to the margins - this was added to the text.

L68: Incorrect period after "m" in 30 – 60 m a-1
**REPLY:** The period was removed.

L98: Surely given the magnitude of previous work done on this glacier, someone has quantified the melt season and you don't have to just assume a start/end date? Or can you use meteorological data to identify when temperatures go above 0 °C on average?
**REPLY:** This assumption was used just to define the period for gathering the satellite images, but in general it corresponds to the meteorological data from the area (station in Petuniabukta close to sea level). The sentence was adjusted "Images covering the ablation season as derived from the meteorological observation in Petuniabukta(15 May – 30

September) with cloud coverage smaller than 20% were extracted and checked for their suitability."

L112: Is there any temperature offset between the two met stations? Do you correct for it?
**REPLY:** The offset is 0.48°C for mean daily temperature during the summer season (warmer conditions in Petuniabukta), the coefficient of correlation is 0.94. We did not correct for it as the offset is rather small and we use the air temperature more as an illustration of the processes. It would be worth correcting when using for modeling purposes etc. We add the information on the offset.

Fig 3b: How is sea ice concentration defined?
**REPLY:** The sea ice cover is compact, a single borderline between open water and sea ice is very clear. We add a more precise definition of the sea ice coverage in the methods section.

L128: The sentence starting with "probably" should be combined with the previous sentence – it is not a complete sentence.
**REPLY:** Thanks for noticing this, it was corrected.

L134: How do you know the water came from snow on the mountains and isn't locally sourced glacier melt?
**REPLY:** You are right, it was meant that it could be ALSO from the snow on the nunataks, the missing "also" was added.

L137: Can you see surface streams in the imagery? That could help you rule out surface connection between the lakes accounting for the synchronous timing. I suspect your latter hypothesis (separated systems responding to a shared forcing) is the most likely.
**REPLY:** We cant see any surface streams on the images actually (which can also mean that these are too small for the 10m resolution). From a field experience we did not observe any surface outflow from a Nordenskiold Lake, for sure there was a huge moulin in the center of the lake connecting the lake to subglacial network. We did not reach the other supraglacial lakes. It is thus hard to make any final conclusion on that.

L139: Clustered how? Spatially or by their behavior? This is also discussion, not results.
**REPLY:** The referenced lakes clustered spatially. Yes, we have the subchapter defined as "Results and Discussion" to facilitate interpretation of our outcomes.

L163-165: You should note that you can also use microwave remote sensing to see through clouds, coarser resolution traditional satellites (e.g., MODIS), or cubesats (e.g., Planet) to get around these cloud & temporal sampling issues.
**REPLY:** Again, a good point. We will elaborate on this and compare our Sentinel-2 dataset to other comparable publicly available satellite remote sensing products. The question is whether these will capture well the sediment plumes.

L166: Subglacial discharge may not make it to the fjord surface if: 1) the flow is hyperpycnal due to high suspended sediment load, or 2) the upwelling discharge plume reaches neutral buoyancy below the surface (e.g., Donald Slater papers). These limitations should be acknowledged in using plumes as your sole proxy for subglacial discharge. (Recurring at L193).
**REPLY:** In general, you are right. In this specific case, we are pretty sure the sediments reach the surface as the outflow from the subglacial channel is near-surface or basically on the surface as the front of the glacier is currently almost landbased. It is quite different from

what is known from massive greenlandic tidewater glaciers for example. We add an explanation to the text.

L189: Citation or more justification needed that the discharge conditions would inhibit vertical water movement.
**REPLY:** We used the bathymetry mapping (Baeten et al. 2010 doi:10.1144/SP344.15) to illustrate the shallow marine environment. The photo from the field to illustrate near surface outflow from the subglacial channels may be used as well.

L198: Strange to say maximum temp is approximately 10 °C and then say 17 °C in the following sentence. Please clarify different metrics or simply statements.
**REPLY:** We agree, the sentence was modified to "The maximum values of approximately 10°C were usually observed during late July in the past years."

L206: This finding could also suggest that subglacial hydrology acts as a flow integrator over these timescales.
**REPLY:** Thanks for this idea, we tried to include that in the text.

L212: "Massive" is subjective, and you don't actually quantify subglacial discharge > reword.
**REPLY:** Sentence was rewritten as: 'Lake drainage is promptly accompanied by significant subglacial runoff expressed through vast meltwater plumes visible on the surface of Adolfbukta near the subglacial drainage outlet in the southern part of the glacier front'

L231: This is stylistic, but I would suggest the conclusion have a "wrap up" sentence instead of ending with this specific example of air temp-plume area relationship.
**REPLY:** Good idea, we will add a couple sentences as a 'wrap up' (as also suggested by Reviewer 2) and put the work in a bigger context.

Review by William Armstrong

---

## Author Comment (AC2)

**Reply to Sierra Melton (RC2)**

Review of "Brief communication: Hydrologic connectivity of a tidewater glacier characterized with Sentinel-2 satellite images – a case study of Nordenskiöldbreen, Svalbard"
Authors: Jan Kavan, Vincent Haagmans

Summary
The authors present an analysis of the changes in supraglacial lake area, meltwater plume area, and sea ice concentration during the summer seasons of 2016-2020 at Nordenskiöldbreen, Svalbard. By manually digitizing these features from Sentinel-2 imagery, they demonstrate that simple remote sensing approaches can yield valuable information on the hydrological system of a tidewater glacier. The data presented in the manuscript is indeed important to understanding Nordenskiöldbreen and is, I believe, worthy of publication. However, the details of the methods are sometimes unclear, and the manuscript is currently lacking in originality and impact. I agree with reviewer William Armstrong that "the study could be strengthened by framing their research around a process-based question about glacier hydrology." There should be more discussion of the significance of this work and new insights; an explanation of what this work contributes to the field of glacial hydrology or our understanding of tidewater glaciers.
Addressing this, clarifying details of the methods, and addressing other comments described below would constitute significant revisions that would help increase the impact of this manuscript.

General Comments
1) There should be more discussion of the limitations of this method, both technical limitations relating to the remote sensing method and environmental limitations in your ability to detect plumes. There is some discussion of technical limitations (temporal resolution and cloud cover) in lines 161-165 (in the Results and Discussion section on meltwater plumes), but this should be expanded on and moved to a different section as it is also a limitation for detecting sea ice and supraglacial lakes. A mention of spatial resolution limitations could also be added (difficulty of observing very small features, if they exist).
**REPLY:** Thanks for this remark. We added a spatial resolution of the Sentinel-2 images to the methods section. We also adjusted the sediment plume detection method to eliminate the manual delimitation - as used in a recently published paper (Kavan et al. 2022 doi: 10.3390/w14121840). The small features such as supraglacial channels can not be observed due to the 10m resolution - this is also discussed now.

2) Regarding the limitation of observing plumes using remote sensing, there needs to be some comment on the possibility of plumes existing without surface expression. The distance a plume rises before reaching neutral buoyancy depends on the discharge volume and rate, fjord stratification, and mélange/sea ice presence/rigidity (Carroll et al., 2015; Slater et al., 2015; De Andrés et al., 2020; Everett et al., 2021). Additionally, the organization of the subglacial hydrologic system determines whether discharging water forms a concentrated plume or not (Melton et al., 2022). A channelized subglacial hydrologic system will result in a discharging plume, whereas a distributed system would result in discharging water to be dispersed along the terminus. Therefore, you cannot conclude that there is no discharge when there is no plume, and you also cannot assume that there is no plume when you do not observe a plume at the surface (in this case there could be a plume at depth that reaches neutral buoyancy before reaching the surface.) You must make it clear that your record of plumes is conservative and be careful about making conclusions about what is happening when a surfacing plume is not visible throughout the manuscript.

**REPLY:** Thanks for this, you are certainly correct about that. We added more detailed description of the glacier front and adjacent marine environment. It was quite obvious for us, but not for the reader. The subglacial channel exits the glacier terminus near surface or almost on the surface as the bedrock is already visible near the subglacial channel. We can add a photo from the field to illustrate it. Regarding this, we are pretty sure to capture all the plumes as they appear directly on the surface. Obviously, we added the information you suggested together with using the references to make the whole issue clear.

3) Details about the methods need to be clarified to ensure that the approach is valid and reproducible. Why was the false-color band combination used? How did you define the edge of sediment plumes? How was sea ice cover fraction calculated? In general, how were all the features (lakes, plumes, and sea ice) defined and identified in the imagery? (This relates to reviewer Armstrong's Major Comment 2.)
**REPLY:** We added a more detailed description to the methods section. A new method for sediment plume detection was used - this helps to eliminate the manual delimitation by adding the sediment concentration proxy - this was tested on another Svalbard site (see (Kavan et al. 2022 doi: 10.3390/w14121840). In general the rest of the features mapped (sea ice extent and supraglacial lakes) are very contrasting, thus relatively easy to delimit.

4) Throughout the manuscript, and especially in the introduction/background, you should consider including additional key references to recent articles about meltwater plumes (e.g., Slater et al., 2018; De Andrés et al., 2020; Cook et al., 2021; Everett et al., 2021; Melton et al., 2022).
**REPLY:** Thanks for providing us with these references, we may have missed some very recent ones. These will make the introduction complete.

5) This is a comment a reviewer once gave me, which also applies here: "As a reader, every time the importance of another study is mentioned by name, it takes the focus away from the amount of work that has gone into this study... A suggested modification would be to summarize the findings of prior research as they relate to this work (instead of who was the author and what they found) and then citing the authors in parentheses at the end of the sentence."
In general, you could aim for more parenthetical citations and less narrative citations.
**REPLY:** Agree, this is very valuable input the authors were not aware of and has been adopted into the manuscript.

Specific Comments
Line 7: Is it appropriate to say that Sentinel-2 images have "high temporal coverage?" Perhaps compared to some other satellite imagery, but you would really need time-lapse imagery to get high enough temporal coverage to see when lakes drain and plumes appear/disappear.
**REPLY:** This was indeed meant as a comparison to other satellite imagery, yet for clarity now has been altered to: 'the usefulness of satellite images from the Sentinel-2 constellation as a simple and accessible tool with daily coverage for studying glacier hydrology in the High Arctic.'

Line 13: The first time you mention Adolfbukta, it would be helpful to clarify that this is the name of the fjord.
**REPLY:** Good point. Has been changed to 'Adolfbukta, the local fjord,'

Lines 20-25: Some of the sentences in the introductory sentences about glacial hydrology need references (the sentences that do not currently have references).
**REPLY:** Will be done. These references must have gone lost in the editing process

Lines 32-34: "Direct measurements of glacier hydrologic processes are usually restricted to short periods and a limited number of sites due to financial, meteorological, and logistical constraints. In response, the indirect study of glacier hydrology and runoff production through remote sensing has gained traction"
This is almost the exact same wording as in the abstract. Re-write either this or the abstract to fix this repetition.
**REPLY:** The wording in the abstract will be slightly altered/reformulated to avoid using almost the exact same wording.

Line 37: "At the same time, the need for expert knowledge to process this abundance of remotely sensed data has been limited." This is unclear. Do you mean that the expert knowledge is limited or that there is no need for it? I expect you mean the expert knowledge is limited, so I would suggest deleting "the need for."
**REPLY:** It is indeed meant that the expert knowledge is limited. Sentence has been reformulated as: 'At the same time, this abundance of remotely sensed data generally requires expert knowledge for further processing, which is not necessarily available to those interested in the rich information captured by the Sentinel-2 satellite constellation'

Lines 40-42: "By quantifying the areal extents of key dynamic features (i.e., evolution of sea ice, supraglacial lakes, meltwater plumes) as proxies of glacier hydrology from Sentinel-2 observations, a simple alternative amidst the outlined constraints is potentially available." Again, word-to-word this is almost the exact same sentence that appears in the abstract. Re-write to avoid repetition.
**REPLY:** The wording in the abstract will be slightly altered/reformulated to avoid using almost the exact same wording.

Lines 44-46: This needs a reference: "Marine areas where such glaciers meet the sea are often considered biological hotspots, directly providing the adjacent marine environment with essential nutrients and other mineral materials."
**REPLY:** A reference has been added: O'Neel et al., S.: Icefield-to-Ocean Linkages across the Northern Pacific Coastal Temperate Rainforest Ecosystem, BioScience, 65, 499–512, https://doi.org/10.1093/biosci/biv027, https://doi.org/10.1093/biosci/biv027, 2015.

Line 55: The Nordenskiöldbreen section reads like a review paper on the glacier. Perhaps it could be consolidated/summarized to include only that information which is directly relevant to this study
**REPLY:** We will look into this and try to consolidate the information as far as it is relevant.

Lines 59-62: You talk about where the glacier calved in the past and allude to the fact that it no longer calves along the entire front. Where does calving happen now? A few sentences describing the current calving behavior would help establish the present-day context. The following sentence touches on this, but it could be elaborated further.
**REPLY:** An additional sentence was added based on our own observations from June 2022 fieldwork: 'In June of 2022, frontal calving was limited to two areas stretching approximately 1 km North and South of Retreat Island along the glacier front.'

Lines 62-63: Move this sentence before the previous sentence to help establish present context before moving on to the future: "Presently, the southern and northern margins and the center of the glacier front (on Retreat Island) are land-based, as derived from satellite imagery through the absence of crevasses."
**REPLY:** Good point, makes it more clear. Has been done.

Paragraph starting at line 78: This paragraph especially reads like a review of all the studies done on the glacier. Consider consolidating this paragraph and using parenthetical citations (at the end of sentences – see General Comment 5).
**REPLY:** Sentences 83-85 were already removed. These focussed on biological processes. Parenthetical citations were used.

Line 90: Describe/define what Petuniabukta is the first time you introduce it.
**REPLY:** Definitely helpful for clarity. Added 'in Petuniabukta, the adjacent bay 10 km to the northwest of the Nordenskiöldbreen glacier front,'

Line 93: Reference for the mean air temperature during the study period? (Or, if this is part of your results, don't include it in this section.)
**REPLY:** Thanks for noticing it, we moved it to the results section.

Figure 1: When I first saw this figure, it was unclear where Terrier and Ferrier Lakes were (especially because it looks like there is no water in Terrier Lake at this time). Is there some way to label the location of the lakes more clearly, either with arrows or drawn outlines?
**REPLY:** The outlines were added.

Line 97: Why was this false-color band combination chosen? This is a NIR-red-green combination (this should be stated for readers unfamiliar with Sentinel bands), which is commonly used for areas covered in vegetation to visualize plant health. Discuss why this falsecolor combination was beneficial to your study.
**REPLY:** This combination was used as the sediment plumes were more contrasting thus easier to delimit. The methods section was adjusted also as a result of using new method for sediment plumes assessment.

Line 98: How was the "assumed ablation season" determined/decided?
**REPLY:** It was based on field meteorological observation (above zero temperatures), this was adjusted in the text.

Line 100: How was it decided that only images from the spring season would be used for sea ice extent?
**REPLY:** We simply covered the transition period from 100% sea ice cover to 0%.

Line 103: How was the boundary of the sediment plume determined? It seems like the visual signature of the plume on the surface of the fjord does not have a clear boundary as the turbid water fades out gradually. Without determining a threshold based on the spectral signature of the turbid water, I'm not sure how a visual estimation of the plume boundary could be standardized.
**REPLY:** In most cases the boundary was rather sharp, but you are right sometimes it was not so evident. To eliminate such effect we determined the sediment plumes with use of NDSSI index and turned it into a measure integrating areal extent and sediment concentration proxy - the method was tested and applied recently (as mentioned above). This should solve this weak point in the areal extent delimitation.

Line 105: This part of the methods is unclear and needs to be elaborated on: "The area of such delimited shapes was then generated automatically in the QGIS software and used for further analyses." What do you mean by "generated automatically?" Do you mean that the areas of the shapes were calculated in QGIS? (If so, say that instead.)
**REPLY:** It is exactly as you wrote - this section was adjusted.

Methods section: For sea ice, how was the cover fraction defined/calculated? What was the total area used to determine the fraction of this total area that was covered in sea ice? Sea ice cover fraction is not mentioned in the methods, but it is plotted in Figure 3.
**REPLY:** Thanks for noticing it. We add a more precise definition of the sea ice coverage in the methods section with description on the calculation.

Figure 2: The label for sediment plume is not in the location where the plume appears to be coming from. An arrow should be used to indicate where the plume is coming from, or the location of the label could be moved.
**REPLY:** We added the arrow.

Figure 3 caption: Should it read "air temperature observed in Petuniabukta and Adventdalen?"
**REPLY:** Indeed, changed to '(a) air temperature observed in Petuniabukta near Nordenskiöldbreen and gap-filled with observations from Adventdalen,' Removed 'Petuniabukta' from the subplot title for Figure 3a'

Lines 128-129: It is suggested that abrupt lake drainage is "probably due to the opening up of subglacial conduits." However, abrupt drainage would be more indicative of opening of a moulin or hydrofracturing through crevasses (e.g., Das et al., 2008; Danielson and Sharp, 2013).
**REPLY:** Fully agree on a second thought. Has been changed to '**their subsequent drainage was rather abrupt which is indicative of moulin opening or hydrofracturing through crevasses (e.g., Das et al., 2008; Danielson and Sharp, 2013).**'

Line 132: Why do you think Ferrier Lake only filled in one year? Do you have a hypothesis for the physical mechanism explaining this?
**REPLY:** There might be some conduits draining the basin blocked/frozen which led to filling the basin this particular year. But we dont have any evidence for this.

Line 134: Seconding the comment by reviewer William Armstrong. Why do you propose that the water is from snowmelt?
**REPLY:** We wanted to point out that there is very likely ALSO snowmelt from the adjacent slopes of the nunatak - obviously also glacier melt. We corrected the sentence.

Lines 141-142: "The filling of two supraglacial lake clusters was shifted towards the peak melt season (July/August), while the third cluster had a similar temporal pattern through its filling during June and draining in late June/July." This level of detail may not be necessary for describing someone else's results from a different glacier.
**REPLY:** We changed this part to point out only the necessary issue - clustering.

Line 147: This paper by Shield et al. is focused on Rink Isbræ, West Greenland. While it is good to mention their findings in the background and potentially other parts of the manuscript, I'm not sure it makes sense to mention one result from Greenland so specifically in your results/discussion section. If comparing your results to those of Schild et al., why not compare them to all the other papers studying plumes and supraglacial lake drainage? This could be done more generally, referencing studies together at the end of a sentence (parenthetical citations) instead of within the sentence (narrative citations). (See General Comment 5 and comment above on parenthetical vs. narrative citations.)
**REPLY:** Thanks for this comment - we used this reference as it was quite close to our study design. We adjusted the text and added some more references.

Figure 4: Why is Long Lake's extent not included in this figure?

**REPLY:** There is no clear explanation or reasoning for this. All monitored extents for the three lakes will be included in the updated version of Figure 4.

Lines 155-156: "The subglacial transport of suspended sediments ceased during September, indicated by the lack of meltwater plumes and hence runoff." This inference is too far-reaching. You can't say it ceased because sediment-laden subglacial water could still be discharging without creating a surface expression visible in satellite imagery. Subglacial water could be discharging in a more dispersed pattern without forming concentrated plumes (indicating distributed rather than channelized subglacial hydrology), or the plume could still exist without rising enough to create a surface expression. (See General Comment 2 and our discussion in Melton et al., 2022.) Even if you did know for sure that there was no discharge of suspended sediment, there could still be subglacial transport of suspended sediments beneath the glacier. More analysis would be needed to support this conclusion (e.g., a time-series of meltwater discharge/runoff).
**REPLY:** As mentioned earlier, we are pretty sure to capture all the plumes as the outflow is near-surface or basically on the surface. But you are right there still might be subglacial transport beneath the glacier just not flowing out of the glacier. We adjusted this.

Lines 156-157: "The observed meltwater plumes originated from the southern part of the glacier front and exhibited a stable spatial pattern." Does this say anything about the spatial network of meltwater drainage through and under the glacier? Simple subglacial flow routing based on hydraulic potential calculations could add significance to the location of the observed plume and even lake locations. If DEMs of the glacier surface and bed are available for your study area, you could consider including this. See Everett et al. (2016), How et al. (2017), and Melton et al. (2022).
**REPLY:** We are not aware of a bed DEM, there were some GPR surveys but mostly in its northern part thus not very useful for the subglacial drainage in its southern part. On the other hand, the photos of the southern frontal part suggest the location of the subglacial channel rather well. We add this to the site description.

Lines 161-165: The limitations of this method should be discussed in a different section because they are limitations for everything (including sea ice concentration and supraglacial lakes), not just the plumes. See General Comment 1.
**REPLY:** We agree, this was added to the text.

Line 172: This needs a reference: "A somewhat more prolonged period of meltwater plume activity (beginning of June to the end of September) was observed in Kangerlussuaq Fjord, West Greenland, for non-tidewater glaciers."
**REPLY:** The correct reference was added: 'McGrath, D., Steffen, K., Overeem, I., Mernild, S., Hasholt, B., & Van Den Broeke, M. (2010). Sediment plumes as a proxy for local ice-sheet runoff in Kangerlussuaq Fjord, West Greenland. *Journal of Glaciology, 56*(199), 813-821. doi:10.3189/002214310794457227'

Lines 188-189: "the meltwater plume outlet is situated close to the sea surface." Isn't it at the base of the glacier, not close to the sea surface?
**REPLY:** The glacier has retreated so far it is starting to shift from tidewater to land-based glacier. The marginal parts (adjacent to coast) are already landbased in both the northern and southern part. As a result the subglacial outlet is really close to sea surface.

Line 192-193: "We argue that spring inflow of meltwater originated probably without contact with the subglacial environment as no visual evidence of sediment inflow was observed." This does not necessarily mean that the meltwater didn't have contact with the glacier bed. The sediment plume may not have been visible during this time if it did not reach the fjord

surface, or subglacial discharge could be dispersed instead of concentrated as a plume. See General Comment 2.
**REPLY:** We tried to make this issue clear by adding more detailed description of the study site - as the subglacial outlet is close to the sea surface we think we are able to capture all the plumes. See previous replies as well.

Lines 194-195: This last sentence of the section seems out of place in the results/discussion section without more discussion of ice velocity and/or including velocity data in your study.
**REPLY:** Good point. Was removed from the manuscript instead.

Lines 201-202: "the set of selected Sentinel-2 images was studied to define of a potential shortterm statistical relation between air temperature and meltwater plume activity." This is confusing wording. You can just say what you did (I think a linear regression between temperature and plume area).
**REPLY:** Was redefined for clarity as: 'To further investigate this observation, linear regression was applied to records of air temperature and meltwater plume activity to study a potential short-term statistical relationship.'

Figure 5 caption: Add a short statement of what the "Theil-Sen estimation" is and a reference.
**REPLY:** added: 'This fitting approach is less sensitive to outliers of sample points by choosing the median of the slopes of all lines through pairs of points (Theil, 1950; Sen, 1968).' However, wouldnt it be more correct to add this in the methods part?

Lines 222-224: "The rather abrupt drainage of the supraglacial lakes at the beginning of the summer season activated the whole hydrologic system of the glacier." I think this might be too far-reaching. There was likely water present in englacial and subglacial networks before the lake drainage which could have been active, even if it were not discharging. I'm not sure you can say that the lake drainage activated the whole hydrologic system.
**REPLY:** The point is that drainage of the lakes indicated that the hydrologic system start to be active and transport the water out of the glacier. In other words the water from the lakes started to flow out through the subglacial network. We tried to rephrase the sentence "The rather abrupt drainage of the supraglacial lakes at the beginning of the summer season indicated that the whole hydrologic system of the glacier started to be active."

Lines 230-231: This last sentence of this manuscript does not belong in the conclusions and is especially out of place as the last sentence of the paper. The manuscript needs another couple of sentences or a paragraph on the broader significance and impact of this work / recommendations for future work.
**REPLY:** Good idea, we will add a couple sentences as a 'wrap up' (as suggested by Reviewer 1) and put the work in a bigger context.

Technical Corrections
**REPLY:** All these technical corrections were adopted in the manuscript and, if necessary, sentences were rewritten/-structured.
Throughout the manuscript, there are several times when "the latter" is used, and it is unclear what this is referring to. The first time I noticed it is in line 13: "… clearly indicates the latter in Adolfbukta at the glacier terminus." I would suggest removing "the latter" here and elsewhere and saying specifically what you are referring to. In most cases this can be fixed by simply changing "the latter" to "this."
Line 42: Unclear use of "the latter"
Line 49: Change "Tidewater glacier is" to "Tidewater glacier termini are" or similar
Line 50: Unclear use of "the latter"
Line 51: Change "Even more so" to "This is even more important" or similar

Line 52: Change "continue" to "continued"

Line 53: Remove "also"

Line 67: Change "has ice" to "with ice"

Lines 83-85: For this sentence, all three references could be at the end of the sentence instead of mixed into the beginning, middle, and end.

Line 101: Written as is, "respectively" is not needed. It could be written: "Respectively, 15, 13, 24, 25, and 24 images were obtained for 2016, 2017, 2018, 2019, and 2020."

Line 102: "(see list of dated images in Supplementary material)": This is labeled as Table 1. You can just say "(Table 1)" or, if it is supposed to be in the Supplementary material, "(Table 1 within the supplementary material)"

Lines 103-105: This is repetitive, saying the same thing in different ways: "manually delimiting the areal extent of studied features (supraglacial lakes, sediment plumes, and sea ice extent). Areal extent of each feature was delimited for each image to obtain the timeseries of features studied." Consolidate these two sentences into one to avoid repetition.

Figure 2 caption: change "showing casing" to "showcasing"

Figure 3: The order of lettering in this figure is a bit confusing. Maybe the letters could increase sequentially down the columns (a,b,c in the first column and d,e,f in the second column) so that the lakes would be d,e,f instead of b,d,f. This would make the caption easier to read.

Figure 3f: The title should indicate both Terrier Lake and Ferrier Lake instead of only Terrier Lake.

Figure 3 caption: There are 2 (e)s in the caption. Delete "(e) Terrier supraglacial lake areal extents"

Line 128: At the beginning of the sentence, change "Probably due to" to "This was probably due to" (or combine the two sentences into one)

Line 139: Change "Other than" to "Different from" or "Unlike"

Line 186: Change "depth's" to "depth"

Line 191: Change "as well" to "also"

Line 198: Change "freezing point" to "melting point"

Line 204: "It is no surprise that a positive correlation was found." This is subjective and unnecessary. This sentence can be deleted.

Lines 204-205: Change "The best correlation was found in the case of aggregated five consecutive days' average air temperature preceding the sediment plumes occurrence" to "The best correlation resulted from averaging the air temperature over the five consecutive days preceding the sediment plumes' occurrences"

Line 210: I suggest changing "onset of snowmelt on the glacier surface" to "onset of surface melting" or similar.

Line 212: Unclear use of "the latter." Change to "This"

Line 213: Same comment as above. Change "The latter" to "This"

Line 215: change "with the air temperature getting down to zero" to "while the air temperature drops to zero"

Line 219: change "consecutive seasons" to "consecutive years" or "consecutive summers" or "consecutive melt seasons"